# Power-Oriented Monitoring of Clock Signals in FPGA Systems for Critical Application

**DOI:** 10.3390/s21030792

**Published:** 2021-01-25

**Authors:** Oleksandr Drozd, Grzegorz Nowakowski, Anatoliy Sachenko, Viktor Antoniuk, Volodymyr Kochan, Myroslav Drozd

**Affiliations:** 1Institute of Computer Systems, Odessa National Polytechnic University, 65044 Odessa, Ukraine; viktor.v.antoniuk@gmail.com (V.A.); myroslav.drozd@opu.ua (M.D.); 2Faculty of Electrical and Computer Engineering, Cracow University of Technology, 31-155 Cracow, Poland; gnowakowski@pk.edu.pl; 3Faculty of Transport, Electrical Engineering and IT, Kazimierz Pulaski University of Technology and Humanities in Radom, 26-600 Radom, Poland; as@wunu.edu.ua; 4Research Institute for Intelligent Computer Systems, West Ukrainian National University, 46027 Ternopil, Ukraine; volodymyr.kochan@gmail.com

**Keywords:** safety-related system, component, FPGA-designing, logical and power-oriented checkability, hidden faults, clock signal, consumed and dissipated power, temperature and current consumption sensors

## Abstract

This paper presents a power-oriented monitoring of clock signals that is designed to avoid synchronization failure in computer systems such as FPGAs. The proposed design reduces power consumption and increases the power-oriented checkability in FPGA systems. These advantages are due to improvements in the evaluation and measurement of corresponding energy parameters. Energy parameter orientation has proved to be a good solution for detecting a synchronization failure that blocks logic monitoring circuits. Key advantages lay in the possibility to detect a synchronization failure hidden in safety-related systems by using traditional online testing that is based on logical checkability. Two main types of power-oriented monitoring are considered: detecting a synchronization failure based on the consumption and the dissipation of power, which uses temperature and current consumption sensors, respectively. The experiments are performed on real FPGA systems with the controlled synchronization disconnection and the use of the computer-aided design (CAD) utility to estimate the decreasing values of the energy parameters. The results demonstrate the limited checkability of FPGA systems when using the thermal monitoring of clock signals and success in monitoring by the consumption current.

## 1. Introduction

It can be argued that the presence of analogies from the natural world has led to the development of the human-created computer world. In this regard, energy is a central aspect of all living (biological) systems. Energy is received from the sun, from volcanoes at the bottom of the ocean, where flora and fauna bloom profusely, and many other natural sources. Computer systems are also powered by energy sources. The thermometer allows us to detect the deviations from the normal thermo state in living systems but also in artificial ones. As such, there are well-known studies on the monitoring of digital circuits using thermal sensors [1,2,3].

In support of robust biological systems, energy balance must be maintained by imposing a coordinated ordering (synchronization) of vital life processes. The synchronization functions in computer systems act much the same way in maintaining component integrity; however, they are much simpler. As such, turning the synchronization circuits off disables the components of a computer system and disrupts its operation without turning off the power. Failures in synchronization of digital circuits lead not only to the creation of a terminal disabling functionality but also to hidden failures, which can block error-free control circuits.

The concept of risk is the primary metric used in evaluating safety-related systems. Risk is determined by the product of two factors: (1) the probability of an accident and (2) the cost of the losses that it can cause. Trends in high-risk facilities are constantly increasing the importance of the second factor. However, reducing the probability of accidents [4,5] will certainly reduce the cost of losses but also result in a more robust safety-related system.

This task falls entirely on information technologies implemented in computer systems, which in safety-related applications are transformed into Safety-Related Systems, for example, safety systems of nuclear power plants [6]. According to international standards, these systems are aimed at solving the complex problem of functional safety: both the safety of the system and the facility under control in order to prevent accidents and reduce losses in case of their occurrence [7,8].

Computer systems perform many functions in various fields of production and consumption, but if they have become as significant in these fields as in accident prevention, for example, in healthcare, communications, or financial domains, it will follow that these fields have also become safety-related applications.

Safety-related applications have their own special significant impact on the synchronization problem. Safety-related systems are designed to operate in both normal and emergency modes. However, in emergency mode, the systems face the problem of insufficient checkability of their components, which causes the problem of hidden faults [9,10]. They can be covertly accumulated in normal mode, including synchronization circuits, and can create a real danger for emergency mode, collapsing fault tolerance of circuitry solutions and functionality of safety-related systems and facilities [11,12].

This paper describes the synchronization problems that take into account hidden faults inherent in safety-related systems and also presents a power-oriented monitoring of clock signals that is designed to avoid synchronization failure in computer systems such as a field-programmable gate array (FPGA). The former is based on the feature of synchronization, manifested in reducing the dynamic component of power consumption and by demonstrating how the power-oriented checkability increases in FPGA systems due to improvements in the evaluation and measurement of their energy parameters.

The main contributions consist in the following: (i) implementation of a power-oriented approach for monitoring the synchronization circuits in safety-related systems to counteract hidden synchronization outages in FPGA components, (ii) experiments conducted with the purpose of researching the thermal checkability of synchronization circuits in FPGA systems, (iii) experimental demonstration of the success in monitoring synchronization circuits by changing the consumption currents. Thus, the main challenge is related to improving the monitoring of synchronization circuits in FPGA components of safety-related systems. The key problem is focused on detecting a hidden shutdown of synchronization circuits in FPGA systems designed to operate in normal and emergency modes, based on a change in power parameters. The effectiveness of monitoring the synchronization circuits of FPGA systems by the energy parameters of the dissipated and consumed power is evaluated for the first time in this paper.

The rest of the paper has the following structure. Section 2 reviews the related works addressed to the development of thermal testability and thermal FPGA monitoring. In addition, the capabilities of modern computer-aided design (CAD) in the evaluation of energy parameters of FPGA systems are shown. Section 3 deals with the increasing problem of hidden faults, aggravating the consequences of synchronization failures, and related aspects of logical and power-oriented circuit checkability. The evolution of the traditionally used logical checkability is presented according to the resource approach. Due to the last one, the logical checkability is limited in the domain of safety-related applications, and there is a need to develop alternative forms, including power-oriented ones. Section 4 shows the results of experiments according to the evaluation of power-oriented checkability for FPGA systems and their monitoring to detect hidden faults in the synchronization circuits. The checkability of the circuits implemented in FPGA systems and the possibilities of their monitoring by the energy parameters of the dissipated and consumed power using temperature and current sensors are studied.

## 2. Related Works

Existing works related to the current problem can be divided into the following four groups: (i) thermal testability and thermal monitoring; (ii) capabilities of modern CAD in the evaluation of energy parameters in FPGA systems; (iii) monitoring of clock signals in FPGA; (iv) a need for developing the power-oriented checkability of circuits in safety-related applications, including monitoring of synchronization circuits in safety-related systems.

Within a *first* group, V. Székely et al. suggested the methodology of design for thermal testability “… to find those chips that operate at higher temperatures than their normal operating temperature; to indicate possible mounting defects; to monitor during the whole lifetime in parallel with normal operation in order to warn of the danger of possible thermal runaway in advance … ” [13]. Yi Ren noted that “an all CMOS (complementary metal-oxide-semiconductor) temperature sensor test is proposed for submicron circuits to detect abnormal temperature changes so as to detect defects on a chip, and increase reliability and service life of devices” [14]. J. Altet and A. Rubio focused on the thermal testing of manufacturing catastrophic defects “to detect hot spots in the integrated circuits” [15].

An overview of the works on thermal testability and thermal monitoring of FPGA systems shows their focus on the overall assessment of thermal regimes. These studies do not analyze the ability to detect a particular type of fault, including synchronization failures.

Within a *second* group, modern CAD systems supporting FPGA-designing include utilities for the preliminary and current evaluation of energy parameters for developed schemes and offer external and internal sensors for their assessment. The utilities enable to set and account the activity of input and internal signals affecting the estimation of the dynamic component for the consumed and dissipated power of FPGA systems [16,17]. Electronics companies offer a wide range of external sensor chips that can monitor both chip temperature [18] and consumption currents for FPGA supply circuits [19]. In addition, some FPGA families have built-in crystal temperature monitoring and tools [20].

Monitoring of FPGA circuits on energy parameters is developed through support from CAD, which already provide utilities and sensors for assessing and measuring not only temperature but also currents characterizing power consumption. We can also note the improvement in sensor accuracy, which enables to enhance circuit monitoring in changing their energy parameters and creates conditions for fault detection with not only catastrophic changes in thermal modes for operation of integrated circuits.

Within a *third* group, C. Metra et al. noted in [21] that the checker of the self-checking register could not reveal a stuck-at fault affecting the clock signal and proposed the method for concurrently checking clock signal correctness in distribution networks of synchronous systems. This method and proposed “self-checking VLSI circuitry that concurrently checks clock signals for permanent and temporary faults which change signal waveforms from those expected from fault-free signals”. Pei Luo and Yunsi Fei proposed to monitor the clock signals and detect glitches in FPGA for opposition to fault injection attacks in cryptographic applications by using a new scheme of comparison of the clock signals with reference clock [22]. G. L. Le et al. presented “a circuit and method herein for monitoring the status of a clock signal. The method includes supplying a pair of clock signals to a clock monitor circuit” for comparison [23].

Overall, we should notice that studies in the area of the clock signal monitoring in FPGA are dedicated to detecting clock glitches using reference clock signals and not the cause of these glitches. Indeed, supplying a clock signal to the register input does not guarantee that the register gets the clock signal because a circuit break can happen after the monitoring point. This loss of the clock signal can be detected by changing the power parameters.

The works of the *fourth* group are considered in the next separate Section 3. Because they not only note what has been done in the field under consideration but also determine the importance of developing energy-oriented approaches. Especially if we take into account new challenges associated with the development of safety-related systems and the limited capabilities of traditional logical checkability in relation to FPGA design.

## 3. Logical and Power-Oriented Checkability

Checkability is widely known in its simplest form—testability, i.e., the suitability of the digital circuit for the development of tests to identify its malfunctions. Testability is structural checkability as it is completely determined by the structure of the circuit [24,25].

During online testing, the checkability of digital circuits also depends on input data and becomes structurally functional. It is advisable to consider the development of circuit checkability in safety-related systems according to the resource approach [26].

Due to this approach, the integration of models, methods and tools combined by the concept of “resources” into the natural world are analyzed, and three levels in the development of resources are determined: replication, diversification and self-sufficiency as the goal of development. Replication is represented in the natural world by integration, which occurs due to a higher birth rate compared with mortality, which is typical of rodents, insects and bacteria. Replication is the simplest form of integration and will always be chosen in the absence of stamping restrictions, i.e., when there are open resource niches: market, technological, environmental and others. At this level, successful development is possible due to increasing productivity.

In today’s computer world, replication is the dominant level of development. Hardware is stamped on the basis of matrix structures, including parallel shifters and adders, iterative array multipliers and dividers [27,28].

Large-sized software modules are stamped and connected to new software products to implement only a small amount of their functions. This negative process of slagging programs is supported by resource niches that are open for the performance and memory capacity of modern computers.

A niche filling process leads to closing resource niches when stamped clones are doomed to extinction. They can only survive if they show their own peculiarities and become individuals, that is, by moving to a higher level of development—diversification. Such a process is observed in green technologies, for example, in mobile systems where memory capacity and performance are dependent on the battery charge limit [29,30].

Safety-related applications stimulate closing of resource niches and moving to the level of diversification, where integration is based on increased trustworthiness, i.e., adequacy to the natural world, including aspects of functional safety. Under these conditions, computer systems rise to the level of diversification and become safety-related systems, diversifying the operating mode by dividing it into normal and emergency ones. The diversification process inherits the input data of digital circuits and their structural and functional checkability, which depends on these data that differ in normal and emergency modes, leading to the problem of hidden faults. It is important to note that this problem is inherent only in safety-related systems. In conventional computers, hidden faults do not create problems as they remain hidden during the full operating mode [31].

Hidden faults can occur not only due to the problem with the fault tolerance function of circuits in a very important emergency mode. This issue is better known for unsuccessful attempts to detect hidden faults while simulating emergency conditions. Unauthorized activation of these modes due to human factors or because of a malfunction has repeatedly led to accidental consequences. The controlled switching on simulation modes is not less dangerous due to the shutdown of emergency protection, which led to the Chernobyl disaster [32,33].

As described above, checkability is logical since it ensures carrying out logical control based on the detection of errors in the calculation results. Logical control is most widely employed in digital circuits, testing them during the pauses and in online testing using actual data [34,35].

The drawback of logical checkability is that it is limited to digital component circuits of modern safety-related systems. These circuits are traditionally based on matrix structures for processing data in parallel codes, which is the main reason for the low structural and functional checkability, especially in conditions of a little change in input data during normal mode. The current dominance of matrix structures requires the development of the concept of circuit checkability by diversifying its forms, among which the power-oriented form seems to be the most promising.

Works on circuit checkability and monitoring in order to detect violation of the thermal mode in integrated circuits due to power dissipation by measuring temperature, including thermal testability studies for safety-critical applications [36], developing online thermal monitoring methods [37], as well as techniques for measuring the temperature of local sections of FPGA microcircuits [38] are known today.

The interest in monitoring circuits due to power dissipation is clarified by the availability of temperature sensors. However, their limited accuracy has become a significant obstacle in the development of this direction. That enables the evaluation of the performance of FPGA systems as a whole with a significant change in energy consumption. Under these conditions, the task of power-oriented fault detection in synchronization circuits was not set.

Assessment of the current state of this issue and possibilities of the circuit power-oriented monitoring in synchronization circuits in the context of improving FPGA designing and its distribution in safety-related applications require experimental studies.

## 4. The Results of Experimental Studies

### 4.1. Experimental Conditions

The power-oriented checkability of circuits in synchronization chains is based on a decrease in power consumption of the dynamic component due to a decrease in the number of clock signal switching. The circuit is suitable for monitoring changes in the energy parameter beyond its values that are possible with proper functioning. During the experiments, lower values of energy parameters are estimated and compared at the correct operation of the investigated circuit and values of energy parameters for this circuit at disconnection of synchronization in bits of its registers.

The experiments evaluated the checkability of the circuits and their monitoring capabilities according to the parameters of the dissipated and consumed power. In the case of dissipated power, a series of experiments proofed the sufficiency of estimates obtained for FPGA systems using CAD utilities. To analyze power consumption, experiments are performed on the evaluation board stand, which enables measuring the consumption current of the built-in target FPGA chip directly. The experiments used the same target chip and CAD in both cases.

The experiments are carried out on the example of *n* bit iterative array multipliers, taking into account the activity of the input signals of the circuit, which was set at the recommended level of 12.5% of the clock frequency. For internal signals, a vectorless estimation was carried out.

Iterative array multipliers contain two *n* bit operand registers and a 2*n* bit product register, which receives clock signals with a frequency of CLK = 115 MHz, which is maximum for *n* = 64. FPGA systems of the multipliers are based on intellectual property (IP) Core LPM_MULT from the Library of parameterized modules [39], and they are implemented on the FPGA Intel Cyclone 10 LP: 10CL025YU256I7G target microcircuit under control of CAD Intel Quartus Prime 20.1 Lite Edition [40,41].

The selected FPGA contains 132 built-in 9 bit multipliers, on which LPM_MULT is implemented. The structure of the 9 bit multiplier (Figure 1) contains the input 9 bit busses of the operands *Data A* and *Data B*, reset *aclr* and synchronization *clock* inputs, as well as the output 18 bit bus of the result *Data Out*. The FPGA systems under investigation are based on built-in 9 bit multipliers.

### 4.2. Investigating the Thermal Checkability and Monitoring the Circuit of FPGA System

Modern CAD systems offer advanced tools for evaluating the energy parameters of FPGA systems, including the PowerPlay power analyzer utility, which enables estimating the dissipated power taking into account the dynamic and static components for the core, as well as the input/output system [41,42].

While evaluating power dissipation, the core is considered as all the structural elements of the FPGA chip. The power dissipation estimate is converted by the utility to the corresponding crystal temperature value.

Before performing calculations of energy parameters, it is necessary to set the predicted values of the input and internal signals activity of the FPGA system in the Power Play power analyzer. The activity or frequency of signal transitions in the system circuit has a decisive influence on the dynamic components of the consumed and dissipated power of the FPGA system.

For input signals, the activity can be estimated as a percentage of the value for the system clock signal or set by the number of transitions per second. The activity value of internal signals can also be set manually, similarly, to input signals, or calculated automatically by the utility based on the activity of inputs taking into account the structure of FPGA system circuit (vector less estimation).

In addition, before evaluating the utility, it is necessary to set the temperature and cooling system values for the system. They include defining the boundaries of the operating temperature range of the *T_J_* crystal, as well as the ways of its determination.

The first option involves setting the supposed fixed value. The second one includes automatic calculation of the crystal temperature by the utility depending on the set parameters of the operating conditions, namely the ambient temperature and the selected cooling system. The lack or the availability of different types of chip cooling sets the corresponding values for thermal resistances between the crystal and the environment.

The results of investigating the FPGA system of the 32 bit iterative array multiplier are indicated in Table 1, including the power dissipation and the corresponding crystal temperature values with the activity of 12.5% for the input and internal signals of the circuit with synchronization disabling in *d* bits operand registers, where *d* = 0, 16, 32, 48.

The total power dissipation *P_D_* is represented in milliwatts by its three components: dynamic *P_DCD_*, static *P_DCS_*, and dissipated power *P_DIO_* of input/output system. These estimates are obtained employing the PowerPlay power analyzer.

Moreover, Table 1 shows the main change in power dissipation in its dynamic component. Synchronization disabling leads to a decrease in this component from 18.61 mW to 7.99 mW, which affects the total power dissipation in its decrease from 166.16 mW to 153.05 mW.

PowerPlay power analyzer calculates the crystal temperature:*T_J_* = *P_D_*·*R_JA_*·10^−3^ + *T_A_*,(1)
where *R_JA_*—crystal–environment thermal resistance, *T_A_*—ambient temperature *T_A_* = 25.0 °C.

*R_JA_* thermal resistance is a constant for the FPGA system; its value depends on the presence or absence of a cooling system. In our experiment, *R_JA_* = 30 is determined by CAD Intel Quartus Prime.

The temperature values *T_J_* calculated according to Formula (1) for the corresponding values of the total power dissipation *P_D_* are indicated in the right column of Table 1.

The circuit is suitable for monitoring synchronization failures in the event of a decrease in the energy parameter below the lower boundary of proper functioning. This boundary is determined by the utility with the smallest, i.e., zero activity of input and internal signals. Zero signal activity reduces the total power dissipation to the level of *P_D.A_*_0_ = 148.83 mW. Then, the lower boundary of the dissipated power can be estimated, taking into account the utility error of ±2.5% [43] as *P_D.MIN_ = P_D.A_*_.0_ (1 − 0.025) = 145.11 mW. In this case, the lower temperature boundary is determined by the Formula (1): *T_J.MIN_* = 0.14511 × 30 + 25 = 29.4 °C, where *P_D_* = *P_D.MIN_*.

Then, the checkability of the circuit makes it possible to detect synchronization failures when the crystal temperature drops below 29.4 °C.

While monitoring the circuit of the FPGA system in order to detect synchronization failures in reducing the crystal temperature, one should also take into account an error of the temperature sensors, the best of which introduce an error of 0.1 °C [44,45].

Therefore, monitoring can only detect failures that decrease the temperature below 29.3 °C.

The results of monitoring (Figure 2) show that when synchronization is disabled in 48 bits of registers (37.5% of the total number of circuit sync inputs), the temperature of the crystal drops from 30.0 °C to 29.6 °C and remains in the operating mode of the temperature values that the sensor can identify with the proper functioning of the circuit. Thus, the activity of the failures in the synchronization circuits is insufficient for its detection by monitoring the crystal temperature.

The results of investigating the FPGA system of the 64 bit iterative array multiplier are illustrated in Table 2, which also contains the values of dissipated power, its components and the corresponding crystal temperature. The last one is obtained using the PowerPlay power analyzer with the same signal activity of 12.5% and synchronization disabling in *d* bits of the operand registers when *d* = 0, 16, 32, …, 96.

Table 2 shows the impact of synchronization disabling on the total power dissipation and its dynamic component by reducing disabling from 189.72 mW to 171.42 mW and from 38.25 mW to 24.99 mW, respectively. The crystal temperature calculated by PowerPlay power analyzer by Formula (1) when *T_A_* = 25.0 °C is indicated in the right column of Table 2.

The lower boundary of the temperature *T_J.MIN_* of proper functioning is determined by the PowerPlay power analyzer due to the total power dissipation *P_D.A_*_0_ = 160.90 mW, calculated when there is zero activity of the input and internal signals taking into account its decrease to the value *P_D.MIN_ = P_D.A_*_.0_ (1 − 0.025) = 156.88 mW and with the utility error of ± 2.5%.

According to Formula (1), the lower temperature boundary is calculated as *T_J.MIN_* = 0.15688 × 30 + 25 = 29.7 °C, and it determines the suitability of the circuit of the FPGA system to control when the crystal temperature drops below 29.7 °C.

Monitoring of the FPGA system is carried out within the framework of the calculated checkability, which is reduced at the same time due to the error of temperature sensors.

The smallest error of 0.1 °C of the most accurate temperature sensors limits the monitoring capabilities in detecting failures to a temperature which is below 29.6 °C.

According to the results of monitoring (Figure 3), synchronization disabling in 96 bits of registers (37.5% of the total number of circuit sync inputs) reduces the temperature of the crystal from 30.7 °C to 30.1 °C, i.e., to a value that relates to the possible sensor readings with the proper functioning of FPGA system.

Thus, both considered examples indicate the impossibility of detecting failures in synchronization disabling even in 37.5% of the sync inputs of the circuit. Thermal monitoring is ineffective due to insufficient thermal checkability of the circuits in relation to synchronization failures and the limited accuracy of modern temperature sensors.

### 4.3. Investigating the Checkability of FPGA System and Its Circuit Monitoring by Current Consumption

Another possibility of power-oriented monitoring of FPGA systems is provided by the circuit checkability according to the parameter of power consumption, estimated by the current consumption [46].

The experiment was conducted using the evaluation board Intel Cyclone 10 LP FPGA evaluation kit [47], which enables to measure of the consumption current of the integrated FPGA chip. To display the measurement results, the evaluation board is connected to a personal computer.

The evaluation board contains the following components:Target Intel Cyclone 10 LP FPGA chip: 10CL025YU256I7G [40];Programmable generator of clock pulses programmable clock generator;Subsystem for measuring the current consumed by the target microcircuit;Toggle switches and push buttons to control the experiment progress.

Software. A formal description of the experimental device circuit, synthesis of the circuit, its placement and tracing in the space of the target FPGA chip was performed in the Intel Quartus Prime 20.1 Lite Edition CAD environment [41].

The target FPGA chip’s consumption current is obtained and visualized in real time using the Power Monitor software utility [47], which is supplied by Intel along with the Intel Cyclone 10 LP FPGA evaluation kit. This utility is installed on a personal computer to display the ongoing consumption current values getting from the current measurement subsystem.

Description of the experimental scheme. The experimental scheme is complicated compared to the scheme for studying the dissipated power due to the need to generate input signals and implemented for *n* = 16, 32, 48 and 64.

For *n* = 16, the circuit (Figure 4) contains the phase-locked loop (PLL) module for generating the main clock signal, the OpUnit subcircuit for generating the multiplier operands, the subcircuit for disabling synchronization, consisting of 6 AND gates, and the investigated multiplier LPM_MULT.

PLL module for generating the main clock signal. A clock signal with a frequency of 100 MHz is fed to the input of the circuit. This signal is formed by the programmable clock generator, which is part of the evaluation board. The phase-locked loop (PLL) module, which is part of the target FPGA chip, increases the clock frequency up to 115 MHz. This frequency is the maximum frequency for the experimental circuit. Four identical clock signals (115 MHz) are generated at the PLL module, and they are the main clock signals of the circuit.

OpUnit sub-circuit for forming the multiplier operands. Data for multiplier operands are generated in their registers composed of 4 bit *Johnson* counters ensured the uniform switching of all operand bits. The *Johnson* counter is implemented on the base of a cyclic shift register and controlled using a *clock* input and a shift *enable* input. The *n* bit operand registers contain *n*/2 *Johnson* counters, i.e., 8, 16, 24, and 32 for *n* = 16, 32, 48, and 64, respectively. Each *Johnson* counter generates a pair of bits for the multiplicand as well as a pair for the same multiplier bits. Each *Johnson* counter bit switches in 4 times less than the clock signal at the *clock* input. The clock signals are applied to *Johnson* counter clock inputs and the trigger that bisects the frequency and feeds the received signals to shift *enable* inputs of the *Johnson* counter. This reduces the switching frequency of the input signals (operand bits) by 8 times, i.e., decreases it to the recommended level of 12.5% of the clock frequency.

The OpUnit subcircuit for *n* = 16 is shown in Figure 5.

Subcircuit for disabling synchronization. To simulate a fault, the synchronization is disabled in six *Johnson* counters, which form the 12 least significant bits of both operands. To disable the synchronization, the 6 two-input AND gates and the 6 control signals are used, which are supplied to the circuit manually using both toggle *switches* and *push* buttons of the evaluation board to stand. These signals disable the synchronization of *Johnson* counters in bits (0, 1); (2, 3); (4, 5); (6, 7); (8, 9); (10, 11), respectively for both operands of the multiplier. This solution was sufficient for *n* = 16, 32, 48. A case *n* = 64 required more bits to disable the synchronization. For this, the circuit contains a shift register, which advances every 80 s the code disabling the synchronization of the next *Johnson* counter.

The multiplier is designed on the basis of standard IP Core LPM_MULT, similar to that used in experiments to estimate the dissipated power. It contains a multiplication circuit and an output register where the multiplication result is placed, as well as operand inputs, and the asynchronous reset input, and the clock input of the output register.

The experiment technique includes the step of preparing the evaluation board to stand and performing the measurements in operation three modes of the experimental scheme:Normal operation (mode 1);Operation at zero activity of information signals (mode 2);Operation under shutdown conditions of the operand register bits (mode 3).

During the preparation phase of the evaluation board stand, the experimental scheme is implemented into the target FPGA using the Intel Quartus Prime CAD by generating a configuration file with the following uploading in FPGA. The Power Monitor utility is activated to obtain and display the ongoing current consumption.

Further, the consumption current per each operating mode of the device was measured sequentially. Measurements per each mode are performed for one minute. The mode sequences are as follows:Normal operation;The zero-activity mode of the input signals, provided by stopping the formation of operands as well as a result of multiplication (the reset signal was supplied to the circuit and held for one minute);The mode of disconnecting the synchronization of *Johnson* counters, which form the bits 0..1, 0..3, 0..5, 0..7, 0..9, 0..11 and 0..11, 0..13, 0..15, 0..17, 0..19, 0..21, 0..22 of both operands for cases *n* = 16, 32, 48 and *n* = 64, respectively.

Results of experiments. The results of the studies are shown in Table 3 for *n* = 16, 32 and 48.

Table 3 shows *I*_16_, *I*_32_ and *I*_48_ values of consumption current in mA for *n* = 16, 32 and 48 at shutdown of synchronization in *d* bits. A case *d* = 0 corresponds to the normal mode. Experimental circuits contain the 4*n* synchronization inputs and determine the percentage of disconnections as *d_n_* = 100 *d*/(4*n*): *d*_16_ = 100 *d*/64, *d*_32_ = 100 *d*/128, *d*_48_ = 100 *d*/256. The zero-activity mode proofed the lowest values of 14.93 mA, 17.36 mA and 20.99 mA, determining the checkability of the circuits below these values for *n* = 16, 32 and 48, respectively. The error of 0.4 mA for the current sensor reduces the specified lowest values to the threshold values: *S*_16_ = 14.53 mA, *S*_32_ = 16.96 mA and *S*_48_ = 20.59 mA, a synchronization fault is detected below those threshold values. The results of experiments show the detection of a fault when the synchronization is disconnected in at least 4 (6.3%), 12 (9.4%) and 20 (10.4%) bits of operands for *n* = 16, 32 and *n* = 48, respectively.

The power monitor utility shows the current consumption graphs for *n* = 16 (Figure 6).

The red line marks the maximum consumption current value that corresponds to the normal operation of the device. The yellow line captures the minimum consumption value that corresponds to the last experimental mode (disabling the synchronization of bits 0.11 in both operands). The green line shows the current consumption values measured at each moment of the experiment. The charts show, from left to right, the level of normal mode (15.74 mA), zero activity of input signals, return to the normal mode, and the sequential sync shutdown mode. In the case of *d*_16_ = 6.3%, the current consumption decreases to the level of 13.72 mA, which is below the threshold *S*_16_. The results of the studies for *n* = 64 are shown in Table 4.

The zero-activity mode determined that the circuit checkability (*n* = 64) is less than 37.94 mA. The last one is reduced by the sensor error to a threshold of *S*_64_ = 37.54 mA, and the monitoring detects a synchronization fault below that threshold. Fault detection occurs when synchronization is disabled at least in 32 (12.5%) bits of operands.

Figure 7 shows the percentage of synchronization outages starting from which and above the monitoring provides the fault detection in circuit synchronization.

For example, for *n* = 16 the sync fault is detected when 6.3% of the sync inputs are disconnected and with a higher percentage of disconnections.

The growing percentage of synchronization outages with a rise of *n* proofs a corresponding decrease in the detection ability of the monitoring. That can be explained by the nature of the dependencies on *n* the number of synchronization inputs and the complexity of the experimental scheme. The number of synchronization inputs grows linearly and increases by 4 times. The complexity of the circuit is mainly determined by the complexity of iterative array multipliers, which grows quadratically and increases by 16 times. Under these conditions, the switching of information signals increases its influence on power consumption to a greater extent compared with clock signals through both functional transitions and glitches [48,49,50].

The experiment was continued for the FIR7531 non-recursive filter, which is a low-pass filter with a cutoff frequency of 32.45 MHz and a gain of 16. Its implementation (fir_filter project) is included in the Intel Quartus Prime 20.1 Lite Edition CAD demo projects [41]. All basic FIR (finite impulse response) filter modules are described in Verilog. The FIR filter multiplier is implemented on the Intellectual Property Core LPM_MULT from the Library of parameterized modules (LPM_MULT library module). The FIR filter processes an 8 bit operand and forms an 8 bit result at the output using the two clock signals: clk and clkx2. The signal clk clocks the main circuit of the FIR filter, and the signal clkx2 clocks the result register.

The experimental circuit, implemented in the evaluation board, contains 32 subcircuits consisting of an FIR filter and an 8 bit *Johnson* counter to form an 8 bit operand, as well as a 3 bit counter that reduces the activity of input signals to the recommended level of 12.5%, and gates to disable sync signals. The zero-activity mode, provided by the reset of all *Johnson* counters, determined the testability of the circuit below 110.19 mA, reduced by the sensor error to a threshold value of *S*_F_ = 109.79 mA. Monitoring detects a synchronization failure below this threshold.

Table 5 shows the results of monitoring the FIR filters, including the number of *d* and percentage of *d*_F_ subcircuits with the disabled synchronization and the current consumption *I*_F_ measured by the sensor in the evaluation board.

The current consumption graphs provided by the Power Monitor utility are shown in Figure 8.

The fault is detected when the synchronization is disabled in *d* = 4 (*d*_F_ = 12.5%) and more subcircuits. The current consumption is reduced to 109.05 mA < *S*_F_.

## 5. Conclusions

The problem of synchronization failures in digital circuits is brought to the attention because of their significant impact on functioning and possible blocking of logical control circuits aimed at detecting failures. This issue is aggravated in safety-related systems, which cannot only increase the price of failures in the conditions of prevention and elimination of accidents but also provide additional conditions for the manifestation of these failures in emergencies.

Experimental studies of power-oriented forms of checkability and monitoring capabilities of circuits for detecting failures in synchronization circuits are carried out using the iterative array multiplier implemented on an FPGA system under the control of CAD Quartus Prime. The energy parameters of the consumed and dissipated power are estimated due to the values of the consumed current of the FPGA system core and the temperature of its crystal, respectively. Failures in the synchronization circuits are introduced by disabling the sync inputs in the bits of the input registers of the iterative array multiplier.

Experiments conducted for studying the FPGA system due to power dissipation are carried out using the CAD utility and showed minor changes in temperature when the synchronization was disabled in 37.5% of the input register bits. This temperature change is completely blocked by the temperature operating range, taking into account the error of the temperature sensors and therefore excludes the possibility of monitoring the FPGA systems in order to detect failures in the register synchronization circuits by measuring the temperature.

An experimental study of the FPGA system due to the consumed power was carried out by measuring the consumption currents at the evaluation board stand and showed the suitability of the iterative array multiplier circuit for monitoring starting with synchronization disabling for 6.3% of the bits of the input registers (*n* = 16). With an increase in operands to *n* = 64, this figure increases to 12.5%, which is associated with the increasing effect of switching information signals (including glitches) compared to clock signals in the iterative array multiplier. An experiment with FIR filters showed the possibility of monitoring, starting with a 12.5% shutdown of sync signals.

Thus, the insufficient thermal checkability of circuits, further reduced by the limited accuracy of modern temperature sensors, does not allow monitoring the FPGA systems to detect failures in the synchronization circuits of registers due to power dissipation. However, this issue can be successfully solved by monitoring circuits due to current consumption, the measurement of which enables to detect of a synchronization failure, starting with 6.3% of disabled sync inputs.

Further research is planned in the direction of expanding the range of investigated schemes and studying the dependence of the results on the features of the initial data in accordance with the requests of customers related to the development of FPGA-based digital instrumentation and control safety systems for nuclear power plants.

## Figures and Tables

**Figure 1 sensors-21-00792-f001:**
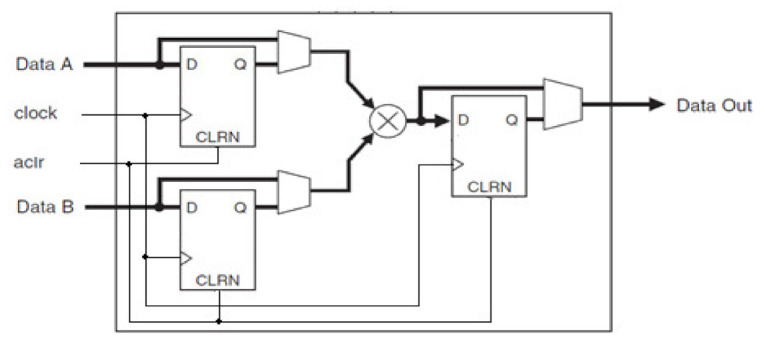
The structure of the built-in 9 bit multiplier.

**Figure 2 sensors-21-00792-f002:**
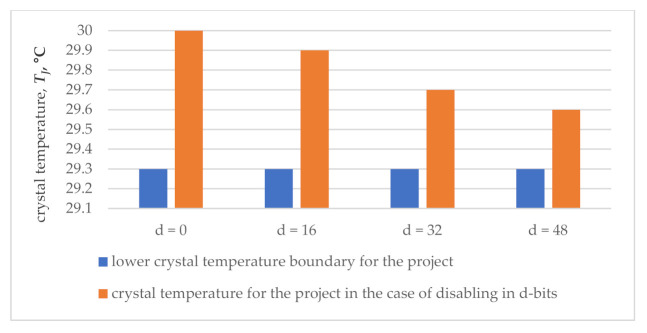
Results of thermal monitoring for 32 bit multiplier.

**Figure 3 sensors-21-00792-f003:**
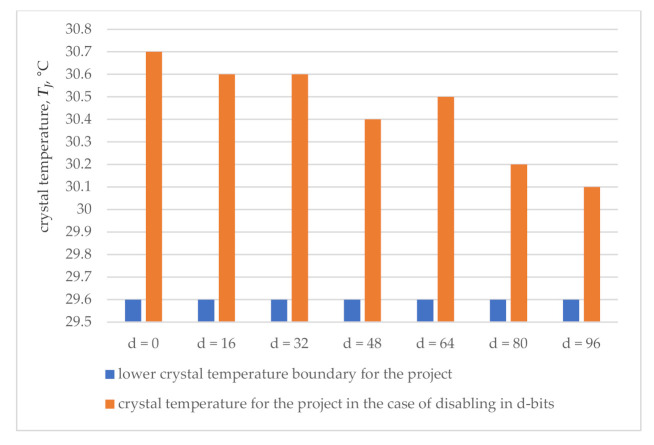
Results of thermal monitoring for 64 bit multiplier.

**Figure 4 sensors-21-00792-f004:**
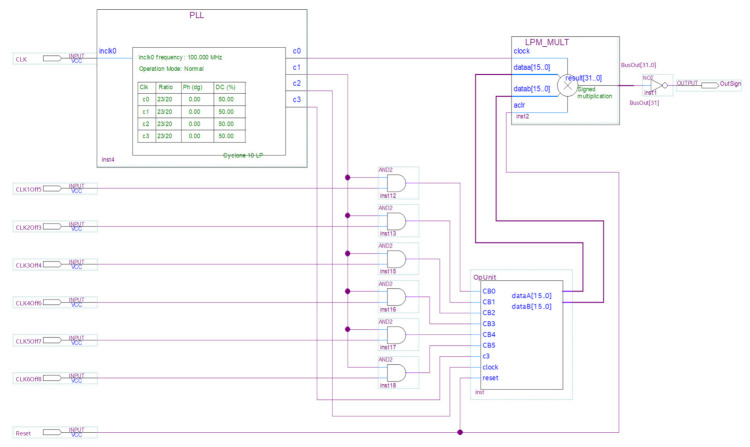
Experimental scheme for *n* = 16.

**Figure 5 sensors-21-00792-f005:**
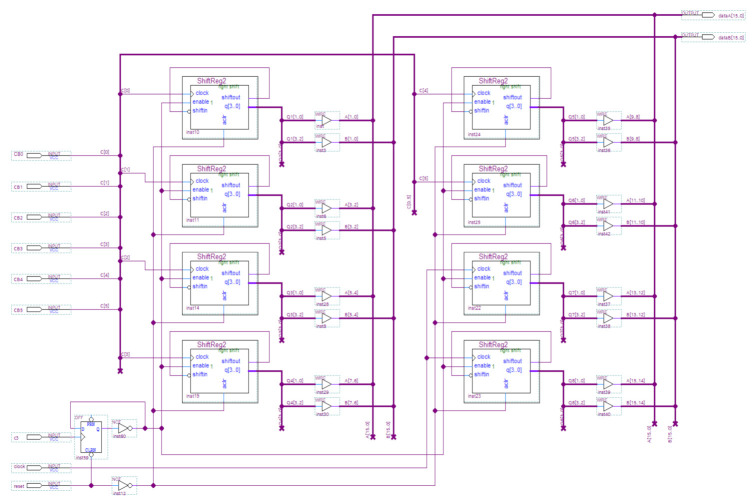
The OpUnit subcircuit for *n* = 16.

**Figure 6 sensors-21-00792-f006:**
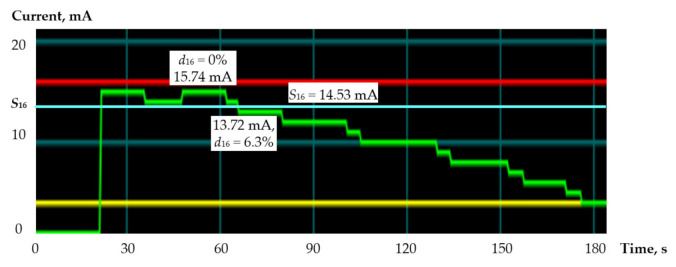
Current consumption charts for *n* = 16.

**Figure 7 sensors-21-00792-f007:**
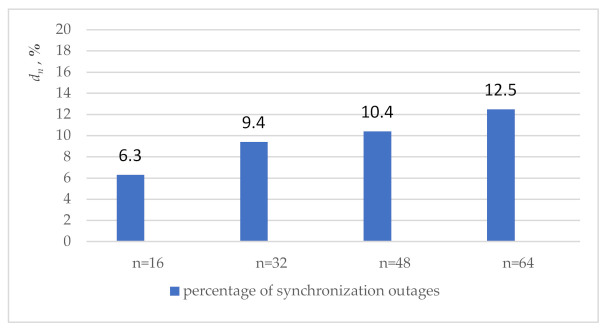
Comparison of results obtained in experiments.

**Figure 8 sensors-21-00792-f008:**
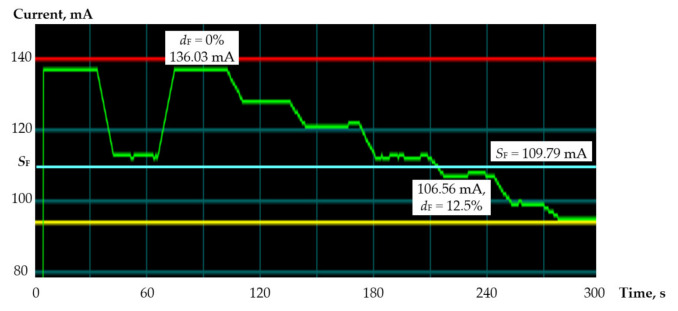
Current consumption charts for FIR filters.

**Table 1 sensors-21-00792-t001:** Dissipated power and crystal temperature of field-programmable gate array (FPGA) system when *n* = 32.

*d*	*d*,%	*P_D_*, mW	*P_DCD_*, mW	*P_DCS_*, mW	*P_DIO_*, mW	*T_J_*, °C
0	0	166.16	18.61	73.25	74.30	30.0
16	12.5	163.96	17.06	73.25	73.65	29.9
32	25.0	157.17	11.33	73.22	72.62	29.7
48	37.5	153.05	7.99	73.21	71.85	29.6

**Table 2 sensors-21-00792-t002:** Dissipated power and temperature of FPGA system chip when *n* = 64.

*d*	*d*,%	*P_D_*, mW	*P_DCD_*, mW	*P_DCS_*, mW	*P_DIO_*, mW	*T_J_*, °C
0	0	189.72	38.25	73.40	78.07	30.7
16	6.3	187.61	37.17	73.39	77.05	30.6
32	12.5	185.15	34.87	73.38	76.90	30.6
48	18.8	180.92	31.22	73.37	76.32	30.4
64	25.0	176.56	28.10	73.36	75.10	30.3
80	31.3	173.87	26.02	73.35	74.51	30.2
96	37.5	171.42	24.99	73.34	73.09	30.1

**Table 3 sensors-21-00792-t003:** Results of experiments for *n* = 16, 32 and 48.

*d*	*d*_16_,%	*I*_16_, mA	*d*_32_,%	*I*_32_, mA	*d*_48_,%	*I*_48_, mA
0	0	15.74	0	20.99	0	29.47
4	6.3	13.72	3.1	18.97	2.1	27.85
8	12.5	12.51	6.3	16.95	4.2	26.24
12	18.8	10.49	9.4	15.34	6.3	24.62
16	25.0	8.88	12.5	13.32	8.3	22.20
20	31.3	6.86	15.6	11.30	10.4	20.18
24	37.5	5.25	18.8	8.88	12.5	18.16

**Table 4 sensors-21-00792-t004:** Results of experiments for *n* = 64.

*d*	*d*_64_,%	*I*_64_, mA	*d*	*d*_64_,%	*I*_64_, mA
0	0	54.49	24	9.4	41.98
4	1.6	53.28	28	10.9	39.36
8	3.1	51.26	32	12.5	37.13
12	4.7	49.84	36	14.1	34.31
16	6.3	47.23	40	15.6	32.29
20	7.8	44.40	44	17.2	30.27

**Table 5 sensors-21-00792-t005:** Results of experiments for FIR (finite impulse response) filters.

*d*	*d*_F_,%	*I*_F_, mA	*d*	*d*_F_,%	*I*_F_, mA
0	0	136.03	4	12.5	106.56
1	3.1	127.95	5	15.6	103.74
2	6.3	121.50	6	18.8	96.07
3	9.4	111.81			

## Data Availability

No new data were created or analyzed in this study. Data sharing is not applicable to this article.

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
