# Peer review of "Power-Oriented Monitoring of Clock Signals in FPGA Systems for Critical Application"

_sensors, 2021, doi:10.3390/s21030792_

Round 1
Reviewer 1 Report
No comments. The work can be published in the present form.
Author Response
Manuscript ID: sensors-1038202
Type of manuscript: Article
Title: Power-Oriented Monitoring of Clock Signals in FPGA Systems for
Critical Application
Authors: Oleksandr Drozd *, Grzegorz Nowakowski, Anatoliy Sachenko, Viktor
Antoniuk, Volodymyr Kochan, Myroslav Drozd
Cover Letter
Authors would like to thank very much for reviewer’s valuable remarks and comments, which helped to improve this paper. All the remarks and comments are taken into account in the revised version of the paper, in particular:
- Are the results clearly presented? Can be improved.
Response: The authors improved the presentation of the results in time charts (Fig. 7 and Fig. 9). The diagrams show the current consumption values below the threshold level when a monitoring detects a synchronization failure.
Sincerely,
Authors 9 January, 2021

Reviewer 2 Report
The paper is generally clearly written. The topic is hot and the methods have some novelty. The authors have made some interesting research summary with overall satisfactory conclusions and analysis.
Comments to improve the paper:
1. The experimental section can be further improved. Some quantitative results can be extended with comparisons against similar approaches. The results can be classified with design motivations.
- Some high level paradigm can be incorporated.
- The analysis and conclusion part needs to be emphasized. The summary with conclusions, future directions, and challenges can be indicated clearly.
- The writing style can be improved.
- The quality of the figures are rather poor. Please revise.
- The major contribution of the paper can be further improved as a journal paper. The current depth and width may be sufficient as a conference paper, but lacks some insightful and complete surroundings considered as a journal paper. The authors can extend the highlights as well as the experimental results when revising the manuscript.
- The benchmarks can be also improved for more complex datasets. More discussions are also favorable for limitations and applicability.
- Some references are quite dated. Please consider more cutting-edge researches when make a background summary and analysis.
Author Response
Manuscript ID: sensors-1038202
Type of manuscript: Article
Title: Power-Oriented Monitoring of Clock Signals in FPGA Systems for
Critical Application
Authors: Oleksandr Drozd *, Grzegorz Nowakowski, Anatoliy Sachenko, Viktor
Antoniuk, Volodymyr Kochan, Myroslav Drozd
Cover Letter
Authors would like to thank very much for reviewer’s valuable remarks and comments, which helped to improve the paper. All the remarks and comments are taken into account in the revised version of the paper, in particular:
- The experimental section can be further improved. Some quantitative results can be extended with comparisons against similar approaches. The results can be classified with design motivations.
Response: The power-oriented monitoring of synchronization disconnections has been investigated for the first time. Therefore, there is no way to compare the results with similar approaches. The results are combined by the study of iterative array multipliers, as prominent representatives of matrix structures in FPGA design, and complemented by the study of FIR filters.
- Some high level paradigm can be incorporated.
Response: As such a paradigm, the authors considered the checkability of FPGA systems, which plays an important role for safety-related applications due to the problem of hidden faults and it is the main limitation of monitoring capabilities.
- The analysis and conclusion part needs to be emphasized. The summary with conclusions, future directions, and challenges can be indicated clearly.
Response: The authors supplemented the conclusion in terms of additional performed experiments and future research in the direction of expanding the nomenclature of circuits studied classes and the dependence of results on the features of the input data.
- The writing style can be improved.
Response: The authors used the help of a native speaker, Prof. Robert Hiromoto (Professor of Computer Science, Center for Advanced Energy Studies 258, University of Idaho, USA, hiromoto@uidaho.edu ) in improving writing style and English.
- The quality of the figures are rather poor. Please revise.
Response: Fig. 1, 2, 5, and 6 were introduced in response to comments from previous reviewers and Academic Editor. To simplify Fig. 5 and 6, the authors replaced the FPGA project of the 32-bit multiplier with the simplest of the considered projects (16-bit multiplier). The authors have limited Fig. 7 showing only time charts that are built by Power Monitor utility
- The major contribution of the paper can be further improved as a journal paper. The current depth and width may be sufficient as a conference paper, but lacks some insightful and complete surroundings considered as a journal paper. The authors can extend the highlights as well as the experimental results when revising the manuscript.
Response: The authors complemented the study of iterative array multipliers by experimenting with FIR filters.
- The benchmarks can be also improved for more complex datasets. More discussions are also favorable for limitations and applicability.
Response: The authors conducted research of power-oriented checkability and monitoring for matrix circuits in safety-related systems developed by the Research and Production Corporation (RPC) Radiy, which is one of the world leaders in FPGA design of safety systems for nuclear power plants (http://www.radiy.com/en/nuclear/about.html). Its products have passed an international audit and are certified for SIL 3. In future the authors plan to expand the types of the investigated circuits and test sequences in accordance with the request of the RPC Radiy.
- Some references are quite dated. Please consider more cutting-edge researches when make a background summary and analysis.
Response: In general, authors tried to build on the recent references, however added links to papers recommended by previous reviewers and Academic Editor. In addition, the sources were used that are closest to the research. Their authors no longer updated publications in this direction.
Sincerely,
Authors 9 January, 2021

Reviewer 3 Report
Dear Authors,
As a reviewer I appreciate the work done and the scientific value it shows. However, in the report that I show below, I indicate improvements that I consider necessary for the work to be published. I hope that the following considerations encourage the authors to deepen their work.
The main objective of this article is to investigate the possibility of detecting synchronization failures in digital systems implemented in FPGAs and distinguish them from inactivity failures in logical input nodes of the circuit. For this, two possible methodologies are analyzed:
- a) FPGA temperature monitoring as an indirect measure of power consumption.
- b) Current monitoring.
The experiments carried out show that option a) is practically ineffective for the objective set (in the case of the design of a multiplier with different word sizes, NxN).
For its part, option b) shows the ability to detect synchronization failures if the number of these occurs in at least a certain percentage of input bits of the multiplier used as an example.
The development of the experiments to support these conclusions seems appropriate. However, there are several issues that need to be addressed by the authors to increase the readability of the article and its quality.
MAJOR ISSUES:
Sections 1 and 3:
These sections really constitute the introduction of the article and should be merged. In addition, they are somewhat confusing and in some parts with a more literary than scientific language. It is recommended, once these sections are merged, to simplify their writing and reduce their size.
Section 4:
First, to validate the conclusions established by the authors it would be necessary to carry out another set of experiments with other types of digital circuits, we propose as an example the design of a CORDIC, or, FIR filters. This is necessary since the results found may be fortuitous and stronger arguments are needed to establish the conclusions stated by the authors. The experiments carried out seem valid and convenient but must be complemented as indicated in the previous paragraph.
On the other hand, to simplify the information of the experiments already carried out, the authors should bear in mind that figures 2, 3 and 4 may be totally unnecessary. It is also recommended that Figures 5 and 6 be modified so that modules appear more easily identifiable with the explanatory text. Figure 7 should also be modified to have a more scientific appearance, showing only the central part of the graph (in black) and not information that misleads the reader at the top and bottom.
Section 5:
The length of this section is totally excessive. In fact, the first part is almost totally superfluous.
MINOR ISSUES:
- I'm not sure that reference [1] is used correctly
- In line 134 delete the word "Reference"
- In line 297 it should be explained in more detail how RJA has been calculated.
- On line 304, it must be explained what the "utility error" is and how the value of + -2.5% was found.
Author Response
Manuscript ID: sensors-1038202
Type of manuscript: Article
Title: Power-Oriented Monitoring of Clock Signals in FPGA Systems for
Critical Application
Authors: Oleksandr Drozd *, Grzegorz Nowakowski, Anatoliy Sachenko, Viktor
Antoniuk, Volodymyr Kochan, Myroslav Drozd
Cover Letter
Authors would like to thank very much for reviewer’s valuable remarks and comments which helped to improve the paper. All the remarks and comments are taken into account in the revised version of the paper, in particular:
MAJOR ISSUES:
Sections 1 and 3:
These sections really constitute the introduction of the article and should be merged. In addition, they are somewhat confusing and in some parts with a more literary than scientific language. It is recommended, once these sections are merged, to simplify their writing and reduce their size.
Response: The authors consider these sections to be important for understanding the problem and justifying the choice of a way to solve it. Moreover, the previous reviewer noted ‘Section 2 present very interested relation to the natural world’.
Section 4:
First, to validate the conclusions established by the authors it would be necessary to carry out another set of experiments with other types of digital circuits, we propose as an example the design of a CORDIC, or, FIR filters. This is necessary since the results found may be fortuitous and stronger arguments are needed to establish the conclusions stated by the authors. The experiments carried out seem valid and convenient but must be complemented as indicated in the previous paragraph.
Response: The authors conducted a FIR filter study using the Evaluation Board and added the results (lines 490-516). The FIR filter has the 8 information inputs, reset and synchronization inputs. Synchronization can only be disabled at the input of the circuit, since there is no access to changes to the circuit itself. Therefore, the experimental circuit contains 32 FIR filter circuits and 8-bit Johnson counters for forming operands, as well as a sequential sync frequency shutdown circuit in d FIR filters, d = 0, 1, …, 6. The zero-activity mode is provided by resetting the Johnson counters. The checkability of the circuit, determined below the zero-activity level of 110.19 mA, is reduced by the sensor error to the threshold value of 109.79 mA, below which the monitoring detects a synchronization failure. The experimental results are shown in Table 5. The Power Monitor utility shows the current consumption graphs in Fig. 9. Fault detection occurs when the synchronization is disabled in 4 FIR filter circuits (12.5%).
On the other hand, to simplify the information of the experiments already carried out, the authors should bear in mind that figures 2, 3 and 4 may be totally unnecessary. It is also recommended that Figures 5 and 6 be modified so that modules appear more easily identifiable with the explanatory text. Figure 7 should also be modified to have a more scientific appearance, showing only the central part of the graph (in black) and not information that misleads the reader at the top and bottom.
Response: Figures 1 and 2 are introduced in response to comments from previous reviewers and Academic Editor. Figures 3 and 4 show the results of thermal monitoring that are observed in the laboratory studies described in the paper. Figures 5 and 6 are also introduced in response to comments from previous reviewers and Academic Editor. The authors have replaced the FPGA project of the 32-bit multiplier with the simplest of the considered projects (16-bit multiplier). Figure 7 shows a report generated by Power Monitor utility of the Evaluation Board stand. The authors limited this report to showing only time charts that are built by the Power Monitor utility.
Section 5:
The length of this section is totally excessive. In fact, the first part is almost totally superfluous.
Response: The authors have shortened a Section 5 by deleting paragraphs 2-4.
MINOR ISSUES:
- I'm not sure that reference [1] is used correctly
Response: According to the authors, references [1-3] reflect the use of a thermometer “to detect the deviations from the normal thermo state in living systems, but also in artificial ones”, in particular, for research human diseases in [1].
- In line 134 delete the word "Reference"
Response: The authors deleted the word "Reference" in the line 134.
- In line 297 it should be explained in more detail how RJA has been calculated.
Response:
The authors explained in line 297 how RJA has been determined. When setting the operating conditions for the project in the Quartus Prime CAD system, the “No heat sink with still air” mode was selected, which corresponds to the real conditions for conducting experiments with the stand. In this case, the thermal resistance of the crystal-environment RJA is determined by the sum of the thermal resistances of the RJC crystal-package and the RCA package-environment, which are constant for the selected microcircuit and equal 7.10 ° C / W and 22.90 ° C / W respectively, and their sum RJA = 30 ° C / W. For example, in Quartus Prime CAD, the path “Assignments” -> “Settings” -> “Operating Settings and Conditions” -> “Temperature” leads to the menu shown below.
This menu for the “No heat sink with still air” mode shows the characteristics of the FPGA design: Junction-to-case: 7.1 ° C / W and Case-to-ambient: 22.90 ° C / W.
- On line 304, it must be explained what the "utility error" is and how the value of + -2.5% was found.
Response: On line 304, the authors made a link to a paper that investigates the PowerPlay Estimation Error for n × n matrix multiplier on Altera FPGAs and added it to References (lines 680-682) as
[43] Renbi, A.; Lindh, L. Power and energy efficiency evaluation for HW and SW implementation of nxn matrix multiplication on Altera FPGAs. Proceedings of the 6th FPGAworld Conference, Stockholm, Sweden, September 2009; pp. 45-51. DOI: https://doi.org/10.1145/1667520.1667526
Therefore, the citing references on the 311, 361, 363, 374, 489 lines have shifted the numbers by one position.
Sincerely,
Authors 9 January, 2021

Round 2
Reviewer 2 Report
No further comments. Revision overall acceptable.Author Response
Manuscript ID: sensors-1038202
Type of manuscript: Article
Title: Power-Oriented Monitoring of Clock Signals in FPGA Systems for
Critical Application
Authors: Oleksandr Drozd *, Grzegorz Nowakowski, Anatoliy Sachenko, Viktor
Antoniuk, Volodymyr Kochan, Myroslav Drozd
Cover Letter
Authors would like to thank very much for reviewer’s valuable remarks and comments, which helped to improve the paper. All the remarks and comments are taken into account in the revised version of the paper, in particular:
Some references are quite dated. Please consider more cutting-edge researches when make a background summary and analysis.
Response: In general, authors tried to build on the recent references. In the same time, we added links to papers recommended by previous reviewers and Academic Editor. In this case, the sources were used that are closest to the research. Nonetheless, the authors have analyzed recent related references once more and updated the links to positions 3, 4, 9, 10, 30.
Sincerely,
Authors 15 January, 2021

Reviewer 3 Report
Dear Authors:
I appreciate the effort made to improve the article.
However, there are some minor actions that must be done before publication.
1.- You must indicate what the filter specifications are.
2.- You must include figures in the X-axis of figures 7 and 9
3.- Figure 2 is totally unnecessary
Best Regards
Author Response
Manuscript ID: sensors-1038202
Type of manuscript: Article
Title: Power-Oriented Monitoring of Clock Signals in FPGA Systems for
Critical Application
Authors: Oleksandr Drozd *, Grzegorz Nowakowski, Anatoliy Sachenko, Viktor
Antoniuk, Volodymyr Kochan, Myroslav Drozd
Cover Letter
Authors would like to thank very much for reviewer’s valuable remarks and comments which helped to improve the paper. All the remarks and comments are taken into account in the revised version of the paper, in particular:
1.- You must indicate what the filter specifications are.
Response: The authors have specified filter characteristics (484, 485 lines).
2.- You must include figures in the X-axis of figures 7 and 9.
Response: The authors have included figures in the X-axis of Figures 6 (7) and 8 (9) (450, 506 lines).
3.- Figure 2 is totally unnecessary.
Response: Authors deleted Figure 2 and corrected numbers for a rest of following Figures.
Sincerely,
Authors 15 January, 2021
